# Modelling of temporal and spatial trends in soil conditions in Finland using HydroBlocks model

Emma-Riikka Kokko<sup>1</sup>, Nathaniel Chaney<sup>2</sup>, Daniel Guyumus<sup>2</sup>, Luiz Bacelar<sup>2</sup>, Laura Torres-Rojas<sup>3</sup>, and Jarkko Okkonen<sup>4</sup>

**Correspondence:** Emma-Riikka Kokko (emma-riikka.kokko@oulu.fi)

Abstract. The changing Arctic climate alters the dynamics of melting and freezing in the ground. An increasing number of frost quakes have been reported in boreal regions such as Finland and Canada, which can cause damage to infrastructure by fracturing roads and built structures. A methodology has been developed to assess frost quakes by estimating thermal stresses in the soil in Oulu, Finland. Information on temporally and spatially varying soil properties, such as soil temperature and soil ice content, is required to calculate thermal stress. Further developing this methodology on a larger scale, over Finland, is challenging due to a lack of in-situ measurements of these parameters with high spatial and temporal coverage. However, they can be simulated using land surface models, one of which is HydroBlocks. Previously, HydroBlocks has been applied in the contiguous United States. The goal of this paper was to configure the model in subarctic and arctic Finland. HydroBlocks' ability to produce accurate snow accumulation and melt approximations, as well as estimate soil temperature and soil water content at different depths in Finland, has not been evaluated before. In addition, maps and time series of soil ice content in Finland at different depths were produced. The snow model (snow water equivalent) and the modeled soil temperatures and soil water contents were compared with the observational data to evaluate the model performance. From the calibrated model, for six observational SWE stations, the average RMSE and KGE were 43 mm and 0.07, respectively. The worst KGE was -0.88, and the best was 0.78. From the calibrated model, for the three observational soil stations, the soil temperature had an average RMSE and KGE of 2.2 °C and 0.66, respectively. The worst KGE was 0.41, and the best was 0.89. For the soil water content, the average RMSE and KGE were first,  $0.15 \frac{vol}{vol}$  and -4.88, and after calibration, they were reduced to  $0.07 \frac{vol}{vol}$  and -0.75, respectively. For the calibrated model, the worst KGE was -2.2, and the best was 0.08. The modelling results emphasize the importance of calibrating the model with local soil hydraulic parameters. The modeling results indicate that outputs from HydroBlocks can generally predict soil conditions in Finland. Furthermore, the obtained soil temperature and soil ice content can be used to calculate thermal stresses in soils and identify frost quake-prone areas regionally across Finland over recent decades, ultimately estimating the risk caused by frost quakes.

<sup>&</sup>lt;sup>1</sup>Oulu Mining School, University of Oulu, Oulu, Finland

<sup>&</sup>lt;sup>2</sup>Environmental and Civil Engineering Department, Duke University, Durham, NC, USA

<sup>&</sup>lt;sup>3</sup>Atmospheric and Oceanic Sciences, Princeton University, Princeton, NJ, USA

<sup>&</sup>lt;sup>4</sup>Geological Survey of Finland, Kokkola, Finland

#### 1 Introduction

Frost quakes, ice-induced small magnitude earthquakes, have caused fracturing of soil and built structures, such as roads and building foundations in the Arctic and Subarctic regions, such as Finland (Okkonen et al., 2020; Puuri, 2024), Canada (Battaglia and Changnon, 2016; Leung et al., 2017), and the USA in New England and states of Illinois and Wisconsin (Hunt, 2024). In the early days of frost quake research in this century, frost quakes have been identified by delineating frost quake suitable conditions from meteorological data (Battaglia and Changnon, 2016). Additionally, the tracking of frostquake events has mostly relied on human observations, for example, by collecting reports made by people on social media (Leung et al., 2017). This is because frost quakes are very localized seismic events, and it is difficult to detect them with the existing seismic network built for earthquake monitoring, unless they occur relatively close to a detector (Okkonen et al., 2020). Fortunately, more recently, frost quakes have also been recorded with a passive seismic network explicitly installed for frost quake detection (Afonin et al., 2024). In Finland, frost quakes have been observed in Talvikangas, Oulu, on  $6^{th}$  of January 2016 (Okkonen et al., 2020) and again a few years later on the same day,  $6^{th}$  of January 2023 (Afonin et al., 2024). However, the second time, they were also recorded with passive seismic stations installed in the area for winter in the ADAPTINFA project of the University of Oulu (Afonin et al., 2024). A similar seismic network was also installed in Tähtelä, Sodankylä, Finland, where frost quakes were recorded during the same winter as well.

Arctic urban environments and their infrastructure (e.g., roads) are at an increased risk of damage (Battaglia and Changnon, 2016; Okkonen et al., 2020; Afonin et al., 2024), due to their direct interaction with the Critical Zone (CZ) of the Earth (Lin, 2010), which comprises soils, quaternary sediments, and the uppermost bedrock, where frost quakes have been detected with an increasing frequency (Battaglia and Changnon, 2016; Leung et al., 2017). Arctic climate is rapidly changing (IPCC, 2023) – For this reason, it is crucial to study the mechanical stability of CZ under the influence of various weather conditions by focusing on the processes in CZ related to seasonal freezing and melting, which further relate to frost quakes. One way to approach this stability issue was presented in previous research (Okkonen et al., 2020), where a method for calculating the thermal stress in soil under different weather conditions was demonstrated. In the study, thermal stresses were estimated in sandy soil in Talvikangas, Oulu, Finland, where frost quakes had been observed on  $6^{th}$  of January 2016 (Okkonen et al., 2020). Thermal stress  $\sigma_{xx}$  in the soil at depth z at time t can be obtained from the spatiotemporal temperature distribution (Timoshenko and Goodier, 1951), by the following equation:

$$\sigma_{xx}(z,t) = -BT(z,t) + \frac{B}{d} \int_{0}^{d} T(z,t)dz + \frac{12(z - \frac{1}{2}d)}{d^{3}} B \int_{0}^{d} T(z,t)(z - \frac{1}{2}d)dz$$
 (1)

Where  $\sigma_{xx}(z,t)$  is thermal stress (Pa),  $B=\frac{E\alpha}{1-\nu}$  is bulk modulus (Pa), E is Young's modulus (Pa),  $\nu$  is Poisson's ratio (-),  $\alpha$  is coefficient of linear expansion (° $C^{-1}$ ), T(z,t) is the soil temperature as a function of depth z, and time t (°C), and d is the thickness of frozen soil layer (m). Furthermore, the thermal stress approach can be applied on a large scale to multiple different soil types across Finland. It would be essential to identify places, where thermal stress accumulated in the uppermost CZ can be released in the form of frost quakes (Okkonen et al., 2020), to prepare protective procedures to safeguard crucial infrastructure.

75

It should be noted that this approach is only 1D. To fully expand this to a larger scale requires developing the thermal stress approach to other dimensions. From the equation (1), it can be seen that, in addition to soil elastic parameters, data on soil temperature (Soil T) and soil ice content (SIC) are required in the calculation of thermal stress, but in situ measurements of these parameters, with high spatial coverage, are rarely available.

However, regional and global land surface data exist that can be used in hydrological and land surface models (Fisher and Koven, 2020) to estimate these variables. In Finland, for hydrological forecasting and research, the Finnish Environment Institute (SYKE) has developed a conceptual WSFS (Watershed Simulation and Forecasting System) hydrological model, and further WSFS-P, a more process-based model, with a two-layer representation of soil in a 1 km<sup>2</sup> grid, which is used to model catchments and especially changes in lakes and rivers across Finland (Vehviläinen and Huttunen, 2001; Menberu et al., 2024). For climate change scenarios and forest-related research, SYKE and the Finnish Meteorological Institute (FMI) have been applying the Max Planck Institute Earth System Models (MPI-ESM) land surface component JSBACH (Thum et al., 2011). In other watershed-scale studies, for example, on peatlands, the HydroGeoSphere (HGS) model has been used (Autio et al., 2023).

For frost quake research, we would benefit from a model that could be used to obtain soil temperature, soil moisture content (SMC), and soil water content (SWC) at different depths with high spatial and temporal resolution. One solution is to use land surface models, one of them being HydroBlocks (Chaney et al., 2016, 2021). In comparison to hydrological models, HydroBlocks is a computationally efficient hyper-resolution model that enables country-scale simulations with a 90 m spatial resolution, basing its calculations on physical processes modelling multiple different surface and subsurface processes at multiple different depths. Previously, HydroBlocks has been applied over the contiguous United States, to estimate SMC, soil T, surface and subsurface runoffs, and to simulate ground water dynamics (Chaney et al., 2016, 2021; Vergopolan et al., 2020, 2021; Guyumus et al., 2025).

In this study HydroBlocks is applied over Finland, located in boreal, Subarctic, and Arctic regions between latitudes of  $59.0^{\circ}$  and  $71.0^{\circ}$  and longitudes of  $20.0^{\circ}$  and  $32.0^{\circ}$ . To utilize the outputs from HydroBlocks in the calculations of thermal stresses and further assess the susceptibility of frost quakes in various types of soils under different climate conditions across Finland, the hydrological simulations must be as reliable as possible. In-situ soil water equivalent (SWE) data from six stations across Finland are used to validate HydroBlocks SWE simulations. Additionally, in-situ SWC and soil T data from three locations in Finland are used to validate HydroBlocks' simulations of the same parameters. Furthermore, from the simulated SMC and SWC, it is possible to determine the SIC at different depths in the soil. Maps and time series are produced for the  $6^{th}$  of January 2016, when frost quakes were observed in Talvikangas, Oulu, Finland (Okkonen et al., 2020), and for the  $6^{th}$  of January 2023, when frost quakes were again observed in the same place (Afonin et al., 2024).

This study utilizes a field-scale-resolving land surface model for the first time to estimate variable fields relevant to approximating thermal stresses - soil T, SMC, and SWC. Ultimately, we anticipate that this will lead to a more accurate and realistic understanding of frost quakes in Arctic regions under the changing climate. This study describes the HydroBlocks modeling workflow and evaluation of the simulations against in-situ measurements. The structure of the paper is as follows:

- 1. The description of HydroBlocks, study area, model setup, and parameterizations.
- 2. Input data used in the model: meteorology and land surface datasets.
  - 3. Model calibration and observational data: Snow water equivalent (SWE) from six stations, and soil station data of soil temperature (Soil T) and soil water content (SWC), from three soil stations.
  - 4. Modeled outputs from HydroBlocks: Maps of SWE, soil T, and SWC, and validation of timeseries against observations of the same parameters.
- 6. Additionally, maps and time series of soil ice content (SIC) were calculated from SWC and soil moisture content (SMC).

## 2 Data & Methods

### 2.1 HydroBlocks

HydroBlocks model is configured for Finland, located in Northern Europe, between latitudes of 59° and 71° and longitudes of 20° and 32°, (Figure 1). HydroBlocks (Chaney et al., 2016, 2021), a land surface model, uses a clustering algorithm to group grid cells of assumed similar hydrological response into Hydrologic Response Units (HRUs) based on datasets of high spatial resolution, such as elevation, soil properties, and land use, that describe the surface spatial heterogeneity. The HRUs are defined by the Hierarchical Multivariate Clustering (HMC) algorithm (Chaney et al., 2021). This approach begins by first splitting a large model domain, for this study, Finland, (Figure 2a), into smaller subdomains, which are shaped as polygons following the natural drainage divide. The basis for delineating the subdomains is a Digital Elevation Model (DEM) used to define the river network and watersheds. The boundaries of the subdomains are adjusted in a way that no watershed is split between polygons, and they only belong to the polygon where they fall the most (Figure 2b).

Figure 1. The location of Finland in Northern Europe. Country extents extracted from the MERIT DEM (Yamazaki et al., 2017) and from the National Land Survey of Finland land cover dataset (MML, 2024), downloaded 7/24. (The coordinate system is WGS84 & units are latitude and longitude.)

Second, the watersheds within each subdomain are grouped into clusters of watersheds by using K-means clustering based on watershed-aggregated covariates (Figure 2b). In this study, the selected covariates include dem, latitude and longitude. In

Figure 2. a) The model domain is divided into 435 subdomains. b) Close up to the highlighted subdomain framed with red color: clusters of watersheds obtained with k = 20. c) The height bands in one cluster of watersheds, framed with a dashed red line, in the previously highlighted subdomain. d) HRUs in the same subdomain after clustering intraband clusters with p = 20.

addition, the empirical cumulative distribution functions (CDF) of HAND values (height above nearest drainage) are calculated for each watershed, which is then set to an average of the HAND empirical CDFs of the watersheds in a cluster of watersheds. This ensures the similarity of the watersheds that are clustered together. Thirdly, from the HAND values, each cluster of watersheds is discretized into height bands, i.e., into a channel and a floodplain component (Figure 2c). The area of each height band increases as they move further away from the channel, and the next height band has n times the area coverage of the previous height band. The last step in defining the HRUs is to cluster pixels within each height band into intra-band clusters based on pixel-based covariates, including latitude, longitude, land cover, and clay content, to account for small-scale heterogeneity (Figure 2d).

Pixels that form the HRUs, do not have to be spatially connected. Still, each HRU is presumed to be homogeneous by taking the average of the meteorological and land data in the pixels that are clustered into the same HRU. Considering the number of HRUs that each subdomain has, at one end of the scope, in theory, a single HRU would be used to describe the whole subdomain. Undoubtedly, with one HRU, it is challenging to capture the real heterogeneity of an extensive land surface area. At the other end of the spectrum, in a fully distributed simulation, each pixel would be its own HRU. However, as a more detailed division is made, more computational power is required. Therefore, a balance between the number of HRUs and computational power is needed to approximate the fully distributed model. The number of HRUs will affect the simulation results. It has been demonstrated that using 1000 HRUs over 610 km² (Chaney et al., 2016) is sufficient to represent the fully distributed solution while maintaining the model's computational efficiency. In more recent work, with the more evolved HydroBlocks (Chaney et al., 2021), it was concluded that a reduced number, 300-350 HRUs, is appropriate to model a 0.25° x 0.25° model domain to approximate the fully distributed solution. However, it is worth mentioning that these are still locationand application-specific scenarios.

160

In total, the whole model domain, Finland, (12° x 12°) is divided into 36 sections (2° x 2°), and each of them has a maximum of 25 subdomains of 0.4°. In total, 435 subdomains where used for the modelling. HydroBlocks' HMC generates 683 689 HRUs using parameters k=20, n=2, p=20. As described in Section (2.1), the model defines the HRUs by delineating the river network and watersheds in all land areas and further clusters pixels hierarchically. This method makes small islands problematic. Hence, Åland, the archipelago of Turku, and the islands along the southern and Western shorelines are excluded from the model.

HydroBlocks uses the vertical column model Noah-MP (Niu et al., 2011) within the HRU framework to represent surface and subsurface processes at a field scale. Each time step, the land surface scheme updates the hydrological states for each HRU; these HRUs interact laterally through subsurface flow at every level of the soil column by computing Darcy's flux and adding it as a divergence term to the vertical solution of Richard's equation in Noah-MP to update the next time step (Chaney et al., 2021). The selected depth of the soil column is 2 m, and the total number of soil layers is 11, located at 2, 5, 10, 20, 30, 40, 50, 60, 80, 100, and 200 cm depth. HydroBlocks is run for the years 2000-2023. The model updates every 20 minutes, but the results are reported on an hourly basis. To evaluate possible trends in the data, the model is run for the same 50-years as a spin-up period. From this procedure, it is concluded that the required spin-up period for the model is between 3 and 5 years. Consequently, in the final simulation, the year 2000 is run 5 times for the model to reach numerical stability, making the physics and dynamics more consistent.

#### 145 2.2 Noah-MP

Noah-MP (Niu et al., 2011) is a more evolved version of the original Noah LSM, with the ability to utilize multiple parametrization. Specifically, Noah-MP offers options for various physical and hydrological processes, allowing the user to choose schemes based on their modeling purposes. Noah-MP simulates various subsurface processes, including recharge, changes in the water table, and base flow. At each time step, the model calculates soil moisture and vertical water flow in unsaturated soils across different soil layers using the one-dimensional Richards equation.

In Noah-MP, the soil column is composed of a user-defined number of layers, and snow cover is modeled with a maximum of three layers, depending on the time-specific snow depth. The snow depth (density) is estimated by taking into consideration how new snow, melting, and refreezing affect the snowpack. The model accounts for the compaction of snow caused by layers above and within a single layer, as well as thermal and melt metamorphism. Considering the ground surface, Noah-MP regards the vegetation canopy as an individual layer, which enables the computation of the surface and canopy temperatures separately. In comparison to bare ground, forested ground intercepts rain and snowfall, and absorbs sunlight differently, consequently affecting the resultant amount of snow on the ground, runoff production, and soil temperature. There is an option for dividing precipitation into rainfall and snowfall. For our purposes, the simplest criterion is chosen, in which all the precipitation comes down as snowfall when the surface air temperature is smaller than the freezing point of water.

For runoff and groundwater, there are options for a simple groundwater scheme (Niu et al., 2007) and a TOPMODEL-based runoff scheme with an equilibrium water table (Niu et al., 2005). Surface and subsurface runoffs are defined as exponential functions that depend on the depth of the water table. With the first option, the saturated hydraulic conductivity decays ex-

ponentially towards the bottom of the soil column, and the model places an aquifer below the soil column. With the second option, the equilibrium water table extends the soil column and performs a water distribution based on the water deficit. The first option is chosen because it reduces the water in the soil column. In addition, there is an option to restrict a zero heat flux from the very bottom of the soil column as a lower boundary condition or, alternatively, use a user-defined value of soil temperature at 8 m depth. The latter is chosen.

Considering more details on frozen soil properties, there is residual water near the soil particles that remain in liquid form even when the soil freezes. The upper limit for this super-cooled liquid water is a function of soil temperature and soil texture class properties, and it defines the maximum amount of liquid water remaining in the frozen soil layer. Koren's iteration (Koren et al., 1999) is chosen, which implements the freezing-point depression equation with an extra term that takes into account also the water around ice particles. Additionally, there are options for the "frozen soil permeability." The chosen option assumes that soil ice reduces the total permeability (Koren et al., 1999).

## 2.3 Study area and input data

195

As input, HydroBlocks requires various types of land surface data, including meteorology, surface elevation, soil thickness, land use, and soil types. The meteorological data is obtained from the Era5 LAND dataset, which is a global climate model (Muñoz-Sabater, 2019). Era5 has been created by combining climate modeling and observational meteorological data via atmospheric forcing. The dataset ranges from 1950 to the present, with a spatial resolution of 9 km and a temporal resolution of 1 hour. The meteorological data used include air temperature, precipitation, air pressure, the Eastward and Northward components of wind, and solar radiation (both short-wave and long-wave).

In Figures 3a & b, the mean temperature and yearly precipitation sum between 2000–2023 is calculated from the Era5 LAND dataset. These maps are similar to yearly statistics from 1991 to 2020 calculated by The Finnish Meteorological Institute, FMI (FMI, 2024a). However, based on visual inspection, the amount of precipitation is higher in Era5 LAND. Although, the years are not the same in both datasets. Further, the Figures 3c & d show time series of air temperature and precipitation for three locations, Sodankylä, Vaala, and Turku created from the ERA5 LAND dataset. It is worth mentioning that in Finland, the yearly temperatures are generally warmer in southern Finland than in northern Finland. Whereas Eastern Finland, located inland, experiences most of the precipitation, and the northernmost regions experience the least. Furthermore, these factors result in varying winter weather conditions across Finland.

A raster map, with a resolution of 20 m of the land use characteristics (Figure 4a), is created from the CORINE Land Cover 2018 dataset obtained from the Finnish Environment Institute, SYKE (SYKE, 2018). Elevation data is provided by the MERIT DEM (Yamazaki et al., 2017), which is a global dataset of terrain elevation with a 90 m spatial resolution (Figure 4c). The soil thickness i.e. Superficial deposit thickness (Figure 4d), was obtained from The Geological Survey of Finland (GTK, 2023). The dataset has a resolution of 250 m, but it has some missing values in the data, especially in Northern Finland, which are interpolated for the model.

Soil texture classes (Figure 4b) are determined with the USDA classification based on the data from the SoilGrids database (Poggio et al., 2021), a global soil model created by combining machine learning, modeling, observations, and environmental

**Figure 3.** a) Mean temperature [°C], b) Mean yearly precipitation [mm], between 2000-2023, and three locations from which there are time series of the corresponding variables. c) Mean temperature [°C] and d) Mean yearly precipitation sum [mm], between 2000-2023 in Sodankylä, Vaala, and Turku. *Original meterological data from (Muñoz-Sabater, 2019)*. (The coordinate system is ETRS-TM35FIN & units are in meters.)

**Figure 4.** a) Raster map of generalized land cover. *Data extracted from SYKE (2018)*. b) Soil texture classes created based on the USDA classification. c) Terrain elevation [m], *Data from (Yamazaki et al., 2017)*. d) Soil thickness [m], *Data from (GTK, 2023)*.

Figure 5. Soilgrids data  $\left[\frac{g}{kg}\right]$  at 0 - 5 cm depth a) sand content, b) silt content, c) clay content, d) organic carbon content. *Data from Poggio et al.* (2021), downloaded 7/24.

covariates. The International Soil Reference and Information Centre has developed the database. The dataset provides the contents of sand, silt, clay, and organic carbon at six different depths: 0-5 cm, 5-15 cm, 15-30 cm, 30-60 cm, 60-100 cm, and 100-200 cm. Figure 5 shows SoilGrids data from the topsoil. This dataset has a resolution of 250 m. SoilGrids' ability to give information not only spatially over Finland but also vertically makes it suitable for modeling purposes in this study. The derived soil texture classes (Figure 4b) divided the Finnish soil into three main categories: sandy loam, loam, and clay loam. "Other" comprehends all the other soil texture classes.

## 2.4 Model calibration and in-situ data

Some of the most critical aspects when simulating soil conditions in Subarctic Finland are the wintertime snow cover, snow melt, the soil types, and the hydrologic properties of those soil types. In Finland, snow accumulates and melts differently in the south, north, west, and east (FMI, 2023) due to different climate conditions. The Finnish Environment Institute (SYKE) measures SWE (Snow line measurements from SYKE (2024)) values across Finland (SYKE, 2022), generally twice a month. For model validation, data from six SWE stations in the cities/municipalities of Sodankylä, Vaala, Kalajoki, Kontiolahti, Konnevesi, and Turku were chosen because they are located at different latitudes and longitudes across Finland. Noah-MP has options for the snow melt curve parameter (MFSNO), which can be set based on the land use type. MFSNO was originally 2.5 everywhere for every land cover type. The values for the MFSNO parameter were optimized based on (He et al., 2019) for each of the land cover types. Based on visual inspection, this slightly adjusted the modeled SWE values.

Additionally, multiple parameters influence the accumulation of moisture or water in the soil profile. In Noah MP, there are values for the characteristics of soil texture classes, i.e., soil types (Niu et al., 2011), such as porosity, saturated soil water diffusivity, saturated soil hydraulic conductivity, quartz content, etc., which affect the SMC (and the soil T). The USDA

classification of soil texture classes, described in the section 2.3, divided Finnish soil into three main categories: sandy loam, loam, and clay loam. However, in reality, Finnish soil is quite different from this, and the soil cover is much more variable, which is evident from the superficial deposits defined by the Geological Survey of Finland (GTK, 2023). Finnish soil is a product of the latest glacial period, and the primary soil type is till, which comprises a diverse mixture of large and small grains of different compositions. Therefore, the USDA texture classes and their hydrological characteristics do not necessarily align with those of Finnish soil. For now, SoilGrids and USDA soil texture classes are still used as the source for soil data due to their spatial and vertical coverage.

In situ data on soil conditions from three soil stations is also available. Two of them are soil stations of the Geological Survey of Finland (GTK) located in Karhinkangas, Kokkola (Hendriksson et al., 2018), and in Juolkusselkä, Sodankylä (Liwata et al., 2014). In Karhinkangas, soil T and SWC were collected at 10 different depths (5, 10, 20, 30, 50, 80, 110, 140, 170, and 200 cm) between 11<sup>th</sup> of September 2011, and 30<sup>th</sup> of May 2015 and in Juolkusselkä at three different depths (20, 40, and 60 cm) between 3<sup>rd</sup> of April 2008 and 11<sup>th</sup> of June 2014. In Karhinkangas, the soil type is sandy soil (Hendriksson et al., 2018), and in Juolkusselkä, it's sandy till (Liwata et al., 2014). In addition, in the ADAPTINFA project, a soil station in Tähtelä, Sodankylä, in northern Finland was installed on 20<sup>th</sup> October 2022. Soil T and SWC have since been collected at five different depths (10, 20, 30, 50, and 80 cm). Additionally, soil samples were taken from the soil pit when the soil station was installed into the ground in Tähtelä, and based on the particle size distributions determined with sieve analysis, the soil type in Tähtelä is medium-grained sand. In Table 1, there is an overview of the differences between soil types/soil texture classes at the locations of the three soil stations from different sources: from in-situ measurements, from the Geological Survey of Finland, and based on the USDA classification.

**Table 1.** Soil types at the locations of the soil stations from different sources.

|                  | Karhinkangas                         | Tähtelä             | Juolkusselkä            |  |
|------------------|--------------------------------------|---------------------|-------------------------|--|
| In situ          | sandy soil <sup>1</sup>              | medium grained sand | sandy till <sup>2</sup> |  |
| GTK <sup>3</sup> | coarse grained soil with peat on top | coarse grained soil | mixed soil (till)       |  |
| USDA             | sandy loam                           | sandy loam          | sandy loam              |  |

Soil types at the locations of the three soil stations from in situ measurements: <sup>1</sup>Okkonen et al. (2020) and <sup>2</sup>Liwata et al. (2014), from the Superficial deposits database of GTK, the Geological Survey of Finland: <sup>3</sup>GTK (2023), and in the HydroBlocks model based on the USDA classification from SoilGrids data.

Further considering specific properties of the soil types at the soil stations, the measured invsitu values of saturated soil hydraulic conductivity in the topsoil at the Karhinkangas soil station can reach  $1.62 \cdot 10^{-3} \frac{m}{s}$  (Hendriksson et al., 2018), while at the Juolkusselkä soil station, they are between  $8.823 \cdot 10^{-7} - 1.261 \cdot 10^{-6} \frac{m}{s}$ . The default values of the texture classes sandy loam and loam are  $5.23 \cdot 10^{-6} \frac{m}{s}$  and  $3.38 \cdot 10^{-6} \frac{m}{s}$ , respectively. In Karhinkangas, the saturated hydraulic conductivities can, therefore, be 1000 times higher than the default values. Values can be this high in some parts of Karhinkangas because it is part of a glacial esker in a groundwater formation area. During the modeling, the saturated hydraulic conductivities of the two most common soil types, sandy loam, and loam, were increased to match the hydraulic conductivity of the Karhinkangas soil

station for one of the experiments. This means more water is flowing through the soil column. This study presents the results from two final models, referred to as Model 1 (M1) and Model 2 (M2). M1 has default values for soil hydraulic conductivity, while in M2, in situ values from the Karhinkangas soil station are used for the two most common soil types, as shown in Table 2.

Table 2. Hydraulic conductivities used in Model 1 & Model 2.

|            | M1 <sup>1</sup>                  | $M2^2$                                          |  |  |
|------------|----------------------------------|-------------------------------------------------|--|--|
| sandy loam | $5.23 \cdot 10^{-6} \frac{m}{s}$ | $1620 \cdot 10^{-6} \frac{m}{s}$                |  |  |
| loam       | $3.38 \cdot 10^{-6} \frac{m}{s}$ | $1.620 \cdot 10^{-6} \frac{\text{m}}{\text{s}}$ |  |  |

Hydraulic conductivities originally in the model  $^1$ Niu et al. (2011), and from measured in situ values at Karhinkangas soil station  $^2$ Hendriksson et al. (2018).

#### 3 Results

Time series of SWE, soil T, and SWC obtained as outputs from HydroBlocks (M1 & M2) were validated against observational data of the same parameters from 6 SWE stations (Sodankylä, Vaala, Kalajoki, Kontiolahti, Konnevesi, and Turku) and three soil stations (Karhinkangas, Tähtelä, and Juolkusselkä). RMSE and KGE were used to evaluate model performance. The SWE values were first validated, because snow on the ground directly impacts the subsurface conditions. The model predicts regionally reasonable values of SWE, but they are higher than the observations indicate. Soil T can be modelled reasonably well, but there is a lag in springtime, which is related to the delay in snow melt time. SWC predictions can be improved with model calibration, which subsequently highlights the importance of using adequate soil data and soil hydrologic parameters.

Maps of SWE and soil T are further analyzed on the dates when frost quakes were observed in Talvikangas, Oulu, 6<sup>th</sup> January 2016, and 2023 (Okkonen et al., 2020; Afonin et al., 2024). In addition, from the SMC, which consists of liquid water and frozen water, and SWC (liquid water), it is possible to calculate the SIC, i.e., the frozen water content as a fraction of SMC. Time series of the SIC in Karhikangas and in Talvikangas are assessed. Further, maps of SIC are also created for the 6<sup>th</sup> of January 2016 and 2023 to evaluate the spatial variation of SIC across Finland and the depth of the frost in the soil on those dates.

## 3.1 Snow water equivalent

Snow cover and snowmelt affect soil temperature and the distribution of water and ice; therefore, simulated SWE is first validated. Figure 6a shows the SWE [mm] from M2, across Finland on 6<sup>th</sup> of January 2016 at 6:00 am, with the locations of the six observational SWE stations. It can be observed that in southern Finland, there is a thin to absent snow cover, with snowpack thickness increasing eastward and northward. In 2023, on the same day, shown in Figure 6b, the snow cover is much more even. Along the Eastern shoreline, the snow cover is the thinnest, while the thickest snow cover can be observed in parts

of Lapland and Eastern Finland. Figure 7 shows the time series at the SWE stations from M1 and M2, and their corresponding RMSE and KGE values are presented in Table 3.

When comparing the timeseries of modeled SWE response to observations of SWE, it can be observed that the model provides reasonable values. Notably, there is a thicker snowpack in Northern Finland (Sodankylä) compared to Southern Finland (Turku). However, the model overall predicts a thicker snowpack than the observations indicate. When comparing the RMSE values between M1 and M2, the difference between the two settings is marginal for each of the analyzed stations. The KGE values between the SWE stations vary a little bit more. In Vaala, Turku, Sodankylä, Kontiolahti, and Kalajoki, satisfactory values are obtained; however, in Konnevesi, the values are unsatisfactory. One noticeable discrepancy in the model is that in springtime, the snow does not melt in an adequate time window. This is shown and discussed in more detail in the following sections.

**Figure 6.** Results from HydroBlocks simulation M2. SWE [mm] on  $6^{th}$  of January at 6:00 am, a) in 2016, b) in 2023 in Finland, with the locations of the six observational SWE stations, which are from SYKE (2024).

**Table 3.** Model statistics: RMSE and KGE values for the six SWE stations in M1 & M2.

|           | Simulation | Sodankylä | Vaala | Kalajoki | Konnevesi | Kontiolahti | Turku |
|-----------|------------|-----------|-------|----------|-----------|-------------|-------|
| DMCE []   | M1         | 49        | 37    | 35       | 58        | 57          | 13    |
| RMSE [mm] | M2         | 52        | 38    | 37       | 59        | 60          | 13    |
| KGE       | M1         | 0.39      | 0.78  | -0.20    | -0.80     | 0.02        | 0.55  |
|           | M2         | 0.33      | 0.77  | -0.28    | -0.88     | -0.07       | 0.56  |

285

**Figure 7.** Results from HydroBlocks simulations: M1 (purple line) and M2 (black line) with observed (colored dots) of SWE [mm] in Finland 2000–2023 at the six observational SWE stations. *Snow line measurements from SYKE, The Finnish Environment Institute (SYKE, 2024).* 

## 3.2 Soil temperature and soil water content

Furthermore, the simulated soil T and SWC are compared with observational data obtained from the three soil stations and Talvikangas, Oulu. There are four maps of the soil T at different depths (5, 20, 40, and 60 cm) from M2 across Finland on 6<sup>th</sup> January 2016 at 6:00 am, in Figure 8 and on 2023, in Figure 9, with the locations of the soil stations. In January 2016, the model predicts colder soil conditions in southern Finland compared to those in northern Finland. In Northern Finland, the soil T does not decrease as much, most likely due to a thick insulating snow cover on the ground at that time, as was shown in the previous section. In Southwestern Finland, the soil temperature is around -3 to -5 °C, even at a 20 cm depth. In January 2023, the snow cover is more even across Finland. At this time, the soil temperature does not decrease below zero in southern Finland, whereas in Northern Finland and Lapland, it is a few degrees below zero. These maps already indicate that the soil in Talvikangas, Oulu, was frozen, especially at the topsoil level, at the beginning of January 2016 and 2023.

The time series of the soil T and SWC for both HydroBlocks experiments corresponding to the locations of the three soil stations are evaluated against observational data from the same depths. Figure 10 shows soil T  $^{\circ}$ C] and SWC  $[\frac{\text{vol}}{\text{vol}}]$ , respectively, at depths of 5, 20, and 50 cm at the Karhinkangas, Kokkola soil station. Figure 11 shows soil T  $[^{\circ}$ C] and SWC  $[\frac{\text{vol}}{\text{vol}}]$ , respectively, at depths of 20, 40, and 60 cm at the Juolkusselkä, Sodankylä soil station. Figure 12 shows soil T  $[^{\circ}$ C] and SWC  $[\frac{\text{vol}}{\text{vol}}]$ , respectively, at depths of 10, 30, and 50 cm at the Tähtelä, Sodankylä soil station. In addition, the air T  $[^{\circ}$ C] and modeled SWE responses [mm] are also shown in each graph. In addition, Tables 4, 5, and 6 show the corresponding RMSE and KGE values for the three soil stations from both of experiments.

**Figure 8.** Result from HydroBlocks simulation M2: Soil T [ $^{\circ}$ C] at a) 5, b) 20, c) 40, and d) 60 cm depth on  $6^{th}$  of January 2016 at 6:00 am in Finland, and the locations of the 3 soil stations and Talvikangas.

Figure 9. Result from HydroBlocks simulation M2: Soil T [ $^{\circ}$ C] at a) 5, b) 20, c) 40, and d) 60 cm depth on  $6^{th}$  of January 2023 at 6:00 am in Finland, and the locations of the 3 soil stations and Talvikangas.

In Karhinkangas, the modeled and observed soil T follow each other relatively well (Figure 10). In addition, the soil temperature follows the air temperature in non-winter months, and the response becomes smoother the deeper it is in the soil. However, in springtime, there is an evident time lag in the model. This is likely connected to the snowpack not melting fast enough. After the snow finally melts, the modeled soil T soon reaches the observational values. Moreover, the winter months (November-February) are the most critical period in the frost quake research because the ground needs to be frozen. The statistical analysis indicates that the RMSE values of soil T do not change too much between models 1 and 2; the RMSE values

300

305

320

decrease at maximum only by 0.4 °C with a mean RMSE over the three depths of 2.2 °C and the mean KGE of 0.77, which indicates outstanding model performances.

Furthermore, when observing the SWC, it can be noted that the statistical analysis indicates that for the M1, the RMSE and KGE values for SWC are not good. Over the three depths, the mean of RMSE is  $0.15 \frac{\text{vol}}{\text{vol}}$ , and the mean of KGE is -6.13. However, by increasing the soil hydraulic conductivity in M2, the modeled SWC reduces closer to the observational values. The statistics are somewhat improved in M2; the mean of RMSE is reduced to  $0.06 \frac{\text{vol}}{\text{vol}}$ , and the mean of KGE is reduced to -0.44. In addition, although the general trend remains one of excess water in the soil, the fluctuations in the observations are also reflected in the modeled responses. This indicates that the model dynamics work, but some of the extra water should flow off as surface runoff. The most notable feature is the stagnant lines in soil water content, which indicate periods when the soil water is frozen, leaving only a residual value of liquid soil moisture. It appears that the model can detect frozen winter conditions.

In Juolkusselkä, similar behavior in the time series (Figure 11) can be observed as in Karhinkangas. In the summertime, the soil temperature matches fairly well; however, there is a time lag in springtime, and similar variations in SWC can be observed between the models and observations. The main difference is that in wintertime, the M1 response is better in soil T than the M2. However, the change in RMSE values is at maximum only 0.2 °C between M1 and M2, with a mean RMSE of 1.82 °C and KGE of 0.64. Considering SWC, for M1, the mean RMSE is 0.13  $\frac{\text{vol}}{\text{vol}}$ , and KGE is -5.6, and for M2, the mean RMSE was 0.06  $\frac{\text{vol}}{\text{vol}}$  and KGE was -0.95. However, the M2 is calibrated with Karhinkangas soil hydraulic conductivity. The in situ soil hydraulic conductivity values in Juolkusselkä are closer to the default values in M1. In Juolkusselkä, the soil type has been defined as sandy till, consisting of fine material with some larger coarse fractions, and the soil station is located on a hill slope.

The time series of the Tähtelä soil station (Figure 12) is much shorter than the previous two. Hence, the temporal variations are more noticeable. The time lag in springtime is observed to be around one month – The snow cover should melt in mid-May rather than June. This also affects the soil moisture as snowmelt water infiltrates into the ground at a later time. The statistical analysis indicates that the RMSE values of soil temperature do not change too much between the M1 & M2, and the RMSE values were only decreased by  $0.2^{\circ}$  with an overall mean of RMSE  $3.0^{\circ}$ C and the mean of KGE 0.6, which indicates a satisfactory model performance. Considering soil water content, for M1, the mean RMSE is  $0.17 \frac{\text{vol}}{\text{vol}}$ , and KGE was -2.9, and for M2, the mean RMSE is  $0.08 \frac{\text{vol}}{\text{vol}}$  and KGE is -0.85.

**Figure 10.** Results from HydroBlocks simulations M1 (purple line), M2 (black line) with observed (pink line) soil T [ $^{\circ}$ C] (on the left) and SWC [vol/vol] (on the right) in Karhinkangas, Kokkola at 3 different depths 5, 20, and 50 cm on  $11^{th}$  of September 2011 -  $30^{th}$  of April 2015. Additionally, there are also the air T [ $^{\circ}$ C] and modeled SWEs [mm] over the same period in both graphs.

**Table 4.** Model statistics: Calculated RMSE and KGE values for the Karhinkangas, Kokkola soil station in M1 and M2 for soil T and SWC at 5, 20, and 50 cm depth.

| Karhinkangas          | Simulation | T5cm | T20cm | T50cm | W5cm  | W20cm | W50cm |
|-----------------------|------------|------|-------|-------|-------|-------|-------|
| RMSE [°C] & [vol/vol] | M1         | 2.7  | 2.4   | 2.0   | 0.14  | 0.15  | 0.17  |
|                       | M2         | 2.3  | 2.0   | 1.7   | 0.06  | 0.06  | 0.07  |
| KGE                   | M1         | 0.65 | 0.7   | 0.78  | -6.4  | -5.9  | -6.1  |
|                       | M2         | 0.76 | 0.81  | 0.89  | -0.45 | 0.08  | -0.94 |

**Figure 11.** Results from HydroBlocks simulations M1 (purple line), M2 (black line) with observed (pink line) soil T [ $^{\circ}$ C] (on the left) and SWC [vol/vol] (on the right) in Juolkusselkä, Sodankylä at 3 different depths 20, 40 and 60 cm on 3<sup>rd</sup> of April 2008 - 11<sup>th</sup> of June 2014. Additionally, there are also the air T [ $^{\circ}$ C] and modeled SWEs [mm] over the same period in both graphs.

**Table 5.** Model statistics: RMSE and KGE values for the Juolkusselkä, Sodankylä soil station in M1 and M2 for soil T and SWC at 20, 40, and 60 cm depth.

| Juolkusselkä          | Simulation | T20cm | T40cm | T60cm | W20cm | W40cm | W60cm |
|-----------------------|------------|-------|-------|-------|-------|-------|-------|
| RMSE [°C] & [vol/vol] | M1         | 2.1   | 1.7   | 1.5   | 0.13  | 0.13  | 0.14  |
|                       | M2         | 2.1   | 1.8   | 1.7   | 0.06  | 0.05  | 0.06  |
| KGE                   | M1         | 0.79  | 0.78  | 0.68  | -3.6  | -5.8  | -7.4  |
|                       | M2         | 0.62  | 0.55  | 0.41  | 0.07  | -0.73 | -2.2  |

**Figure 12.** Results from HydroBlocks simulations M1 (purple line), M2 (black line) with observed (pink line) soil T [ $^{\circ}$ C] (on the left) and SWC [vol/vol] (on the right) in Tähtelä, Sodankylä at 3 different depths 10, 30, and 80 cm on  $20^{th}$  of October 2022 -  $31^{th}$  of December 2023. Additionally, there are also the air T [ $^{\circ}$ C] and modeled SWEs [mm] over the same period in both graphs.

**Table 6.** Model statistics: Calculated RMSE and KGE values for the Tähtelä, Sodankylä soil station in M1 and M2 for soil T and SWC at 10, 30, and 80 cm.

| Tähtelä               | Model | T10cm | T30cm | T80cm | W10cm | W30cm | W80cm |
|-----------------------|-------|-------|-------|-------|-------|-------|-------|
| DMCE [0C] % [1/1]     | M1    | 3.6   | 3.2   | 2.2   | 0.14  | 0.19  | 0.18  |
| RMSE [°C] & [vol/vol] | M2    | 3.4   | 3.0   | 2.1   | 0.06  | 0.10  | 0.09  |
| KGE                   | M1    | 0.49  | 0.53  | 0.68  | -0.73 | -4.1  | -3.9  |
|                       | M2    | 0.55  | 0.6   | 0.73  | 0.15  | -1.2  | -1.5  |

335

## 325 3.3 Soil ice content

From the total SMC and SWC, it is possible to calculate the amount of ice in the soil, i.e., what percentage of the total soil moisture is frozen. Because frost quakes were observed in Talvikangas, Oulu, Finland, on  $6^{th}$  January 2016 and 2023, maps of soil ice content are created for both dates. Figure 13 shows a map of the SIC [%] (ice vol/SMC vol) across Finland at 5, 20, 40, and 60 cm depth from M2 on  $6^{th}$  January 2016 at 6:00 am and Figure 14 on  $6^{th}$  of January 2023 at 6:00 am. For the year 2016, the model predicts relatively high SIC in southern and western Finland, as well as in the northern part of Lapland, where around 60% of the soil water in the topsoil is frozen. Elsewhere, 20-40% of soil water is frozen, and in North-Eastern Finland, the SIC is near zero at 5 cm depth. Moving deeper into the soil, the SIC decreases. In southwestern Finland, the frozen soil ice layer is thicker, and in some places, it reaches a depth of 40 cm. For the year 2023, the model predicts somewhat similar SIC, except for southern Finland, where soil moisture is entirely in liquid form.

**Figure 13.** Result from HydroBlocks simulation M2: SIC [%] (ice volume/SMC volume) at a) 5, b) 20, c) 40, and d) 60 cm depth on  $6^{th}$  of January 2016 at 6:00 am in Finland, and the locations of the 3 soil stations and Talvikangas.

In addition, a time series of SIC in Karhinkangas is obtained, Figure 15. It shows the air T [°C], the modeled SWEs [mm], and the modeled SIC [%] at the depths of 5, 20, and 50 cm at the Karhinkangas, Kokkola soil station from M1 and M2. It can be noticed that both models predict that the soil is frozen in winter times. Reasonably, the topsoil has the most ice, and ice depth differs between years. When observing the air temperature, it can be noted that in 2013, the air temperature was below zero more frequently than in 2012 and 2014, which results in a deeper ice layer that also lasts longer. Considering Talvikangas, in 2016, the soil has ice to at least 20 cm depth, and in 2023, to 40 cm depth. To be able to observe SIC in Talvikangas in more detail, additionally, time series' for winters 2015-2016 and 2022-2023 are created between 1<sup>st</sup> November and 31<sup>st</sup> March, which is also shown in Figure 15. There are the air T [°C] for both years, the modeled SWEs [mm], and the modeled SIC [%], at the depths of 5, 20, and 50 cm from both models, and the red line marks the 6<sup>th</sup> of January. M2 estimates that in 2016 and

**Figure 14.** Result from HydroBlocks simulation M2: SIC [%] (ice volume/SMC volume) at a) 5, b) 20, c) 40, and d) 60 cm depth on  $6^{th}$  of January 2023 at 6:00 am in Finland, and the locations of the 3 soil stations and Talvikangas.

2023, the soil is frozen from the topsoil, but the ice does not reach very deep, only 20 cm, where there is around 15% ice. M1 estimates that ice reaches deeper, even to a 50 cm depth, for both 2016 and 2023, with around 40% ice.

## 4 Discussion

The modeling shows that HydroBlocks can generally predict snow and soil conditions in Finland. Over 24 years, the modeled SWEs indicate more snow in Lapland and less snow in Southern Finland, which is in agreement with the observations. However, the model overall anticipates a thicker snowpack than the observations show. This could be due to the options in Noah-MP, specifically how precipitation is divided into rainfall and snowfall. Consequently, a comparison is performed between Era5Land and one meteorological station of the Finnish Meteorological Institute, FMI, in Tähtelä. From Figure 16, it can be noted that the monthly mean air T between FMI and Era5land matches pretty well. However, there is much more variation in monthly precipitation sum; generally, Era5land has more precipitation than FMI. However, this does not explain the difference by itself.

On the day frost quakes were observed, in Oulu, Finland in 2016 and 2023 (Okkonen et al., 2020; Afonin et al., 2024), our modeling results show that soil is frozen at least to 5 cm depth (Figure 8). On 6th of January 2016 Southern and South-Western Finland. In 2016, in southern Finland, thin snow cover, enabled soil freezing, whereas in Lapland, thicker, insulating snow cover prevented soil temperature from decreasing. In 2023, the snow cover was more evenly distributed across Finland, except for southern Finland, where the soil was frozen throughout the topsoil.

When evaluating the freezing and warming of soil, it was noticed that in springtime, there is an evident time lag in the model. Soil temperature does not increase in an adequate time window, which is likely due to snow melting slower than it does in reality. This could also be connected to the thicker snowpack in springtime before melting starts, i.e., it takes longer

**Figure 15.** Results from HydroBlocks simulations M1 (purple line) and M2 (black line) of SIC [%] in Karhinkangas, Kokkola (on the left) at 3 different depths 5, 20, and 50 cm between  $1^{st}$  of September 2011 -  $30^{th}$  of 2015, and in Talvikangas, Oulu (on the right) at 3 different depths 5, 20, and 50 cm between  $1^{st}$  of November of 2015 -  $1^{st}$  of April 2016 and  $1^{st}$  of November of 2022 -  $1^{st}$  of April 2023. Additionally, there are also the air T [°C] and modeled SWEs [mm] over the same period in both graphs, and January  $6^{th}$  is marked with a red line.

for the snow to melt. The model response is better during the winter months (November-February), which is the period of most significant interest for our research, as the ground needs to be frozen for frost quakes to occur. With the application of HydroBlocks on a regional scale over Finland, the biggest concern is related to the evaluation of the SMC (and, further, the SIC), which is highly dependent on site characteristics and can change significantly within a small spatial window. When observing the behavior of soil moisture, it is essential to note that the model dynamics are functioning correctly, i.e., the fluctuations in soil moisture are accurately captured, even though the level of moisture in the soil is higher than the observations indicate.

One reason that the SWC level does not match is very likely connected to the fact that the USDA soil texture class division and given values for the soil characteristics have been developed for USA soil (Saxton and Rawls, 2006) and, therefore, cannot comprehend the differences in Finnish soil, for example, glacial till. In future work, it would be beneficial to consider using more precise soil data, such as that from the Geological Survey of Finland or from the Finnish Environment Institute. Param-

**Figure 16.** Comparison between monthly air T and monthly precipitation sum in Era5Land and FMI datasets. *Data from Muñoz-Sabater* (2019) and FMI (2024b).

eters controlling soil moisture need to be calibrated spatially to obtain more accurate regional modeling results. Additionally, considering the actual road conditions, snow is cleared away regularly. In the model, this is not taken into account. Therefore, in reality, the temperatures may be even lower without an insulating snow cover. Although it is possible to have frost quakes even with snow on the ground.

#### 5 Conclusions

Main conclusion are as follows:

- 1. Based on meteorological and land surface data (soil types, land use, elevation, and soil thickness), HydroBlocks can generally predict, especially during wintertime, snow and soil conditions in Finland at 90 m spatial and 1 hour temporal resolution, with some uncertainty.
- 2. Modelled SWEs give spatially reasonable values. However, the model predicts a somewhat thicker snow cover than the observations indicate. In springtime, there is a delay of around one month in the snowmelt process, which affects the calculated model statistics. The average (from both models) of RMSE across the six sites was 42 mm, and KGE was 0.10. The best KGE value was 0.78 (M1) in Vaala, and the worst was -0.88 in Konnevesi (M2).

405

410

- 385 3. Modeled soil T and observational values match pretty well. In springtime, there is a time lag in the model, most likely due to snow cover not melting in an adequate time window. The average RMSE across the three sites was 2.4 °C (M1) and 2.2 °C (M2), and KGE was 0.68 (M1) and 0.66 (M2), indicating an outstanding model performance.
  - 4. SWC: The average RMSE across the three sites was 0.15  $\frac{\text{vol}}{\text{vol}}$  (M1) and 0.07  $\frac{\text{vol}}{\text{vol}}$  (M2), and the KGE was -4.9 (M1) and -0.75 (M2), indicating poor model performance. Using a measured value of soil hydraulic conductivity did improve the model performance. In the future, to model soil moisture most accurately, Finnish soil properties should be considered, for example, by utilizing more detailed soil data from the Geological Survey of Finland or from the Finnish Environmental Institute.
  - 5. Additionally, SIC can be calculated from the SMC and SWC, which are obtained as outputs from HydroBlocks. SIC gives estimation on the frost depth and its spatial coverage across Finland.
- 6. Furthermore, the outputs from HydroBlocks, soil T and SIC (from SMC and SWC) can be used to calculate thermal stresses in soils across Finland and to estimate the risk of frost quakes.
  - 7. Moreover, HydroBlocks's ability to produce spatially and temporally varying fields describing surface and subsurface soil conditions has the potential in the future to use the modeled outputs in other environmental applications as well.

Code and data availability. Data files, code for model simulations, and observational data are available at Kokko, E.-R., & Chaney, N. (2025). Supporting Dataset Modelling of temporal and spatial trends in soil conditions in Finland using HydroBlocks model [Data set].

Zenodo. https://doi.org/10.5281/zenodo.16601663. Additional material can be requested from the authors.

Author contributions. ERK downloaded and preprocessed the data for Finland, implemented HydroBlocks over Finland, ran the simulations and analysis, and prepared the first draft of the manuscript. NC, LTR, and DG helped set up the model, fixed all challenges encountered with the code, and provided incredibly valuable feedback over the process of modeling and analysis. LB helped with ERA5 data download and preprocessing. JO provided assistance and feedback, especially with the Finnish observational data and Finnish soil characteristics. All authors contributed to the revision and writing of the final version of the paper.

Acknowledgements. This study was a part of the ADAPTINFA (Urban environment and climate in the Arctic: Data-driven intelligence approach to multihazard mitigation) project that has received funding from the European Union – NextGenerationEU instrument and is funded by the Research Council of Finland under grant number 348802 for 2022-2024. Personal funding (for Emma-Riikka Kokko) for the year 2025 from the Vilho, Yrjö, and Kalle Väisälä Foundation of the Finnish Academy of Science and Letters (decision 15<sup>th</sup> of November 2024).

The research visit to Duke University (March 2023 - September 2023) in North Carolina, USA, was funded by a travel grant from the University of Oulu Graduate School (decision  $19^{th}$  of December 2022). Very special thanks to the researchers at Duke University who made the memorable research visit possible and who offered modelling resources and valuable assistance with HydroBlocks and the writing of this paper.

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
