# Peer review of "Modelling of temporal and spatial trends in soil conditions in Finland using HydroBlocks model"

_EGUsphere, 2025_

## Author Comment (AC1)

Thank you to the reviewer for your feedback and suggestions. Please find the response to your comments in blue.

**Reviewer #1**

The study applies the HydroBlocks model to simulate soil conditions in Finland, and the technical implementation appears thorough. However, I have significant concerns regarding the scientific novelty, methodological clarity, and overall focus of the paper. In its current form, the manuscript does not meet the standards for publication in this journal.

Below are my specific comments:

> 1. Lack of Clear Innovation and Advantage. The study appears to be a regional validation of the HydroBlocks model over Finland. However, the manuscript does not clearly articulate what specific advantages HydroBlocks offers compared to existing reanalysis products or other high-resolution models. Without a clear comparative analysis against established datasets (e.g., ERA5, satellite-derived SWE, in situ observations), it is difficult to assess the added value of this work.

Thank you for this valuable comment, we think this aspect of the paper needs to be improved. In the introduction, as a justification of using HydroBlocks, there is comparison with HydroBlocks to some other models used in research in Finland, for which are used only at a catchment and/or watershed scale such as Watershed Simulation and Forecasting System (WSFS), and HydroGeoSphere (HGS) models. Advantages with HydroBlocks are that it offers us the possibility to model multiple variables computationally efficiently but at a high-resolution scale at different depths across the whole country, and the model could possibly be used in other applications as well. We are presenting a hyper resolution simulation that can be validated with local observations of processes that take place in a field scale resolution.

In addition, in our paper, in the discussion sections, there is also discussion related to the possible problems with ERA5 Land air temperature and precipitation (smoothing of very low temperature and very high temperatures and over estimating precipitation) used global models, as well as differences between Finnish soil and with the Soil Grids soil data. The problem with global models, such as SWAT hydrological model, is the usage of synthetic soil representation, which lacks sufficient presentation of Finnish soil (such as till), which is a product of the latest glacial period. This issue was also brought up in our paper with our simulations. However, HydroBlocks offers us the opportunity to use different datasets for meteorology and soil data to improve the simulations. To strengthen our novelty, we suggest that in addition to presenting HydroBlocks model results for Finland using SoilGirds data (validated with in-situ observations), we would add to this paper HydroBlocks simulations using local soil data provided by the Geological Survey of Finland. This improved presentation of soil would highlight the advantages against other models.

In model validation, we used in-situ observations from six snow water equivalent stations at different latitudes and longitudes across Finland, and for soil temperature and soil water content, we had data from three soil stations. These in-situ observations are more reliable than satellite observations. Satellite-derived SWE could turn out to be valuable to "combine"/ use together with HydroBlocks model in the future, but just as a single dataset, we feel it would not increase the value in the validation.

> 1. Misdirected Introduction Focus. The introduction extensively discusses frost quakes, yet the manuscript does not directly address this phenomenon in the results or discussion. If the goal is to model soil conditions relevant to frost quakes, the connection should be

explicitly demonstrated. Otherwise, the lengthy background on frost quakes seems disconnected from the actual content.

This is a valuable comment, and we agree that the introduction focuses too much on frost quakes, and a shorter description of the phenomena is sufficient for this paper. The motivation of this work is to, in the future, be able to calculate thermal stress across Finland, but because it is a huge work, it will be done in a separate paper. The connection between frost quakes and soil conditions is achieved by introducing the thermal stress equation, which contains the soil temperature and frost depth that can be obtained as outputs from HydroBlocks. The goal of our paper is only to model soil conditions relevant to frost quakes; soil temperature and soil ice content, and for this reason some background on frost quakes and frost quake research is needed, but it will be made shorter.

1. Insufficient Model Performance Evaluation. While the authors note that HydroBlocks performs well in some respects, several KGE values are negative or low. For hydrological modeling, KGE > 0.5 is often considered acceptable for streamflow, but for other variables (e.g., soil moisture, temperature), benchmarks are less clear. The manuscript should include a systematic comparison with independent observational or reanalysis products (e.g., satellite SWE, soil moisture from SMAP or ERA5) to objectively demonstrate model superiority.

We agree that some of the values describing the model's performance, especially with soil water content, are not satisfactory. However, we have also discussed some possible reasons for this; the model's ability to describe Finnish soil is challenging due to the presence of glacial deposits such as till formations. In addition, as discussed in the discussion but will be expanded, for a model that covers the whole country, it is challenging to calibrate it to precisely represent each pixel with relatively low number of in-situ soil parameters. Further, soil water content is quite location specific and can change within a small spatial window. We also think that for model validation, satellite observations would not add value since we already have the in-situ observations. Additionally, we are mostly interested of the times when the soil is frozen, which the model can accomplish. The most important soil condition for us is the soil temperature, which the model can simulate well.

1. Unclear Purpose of Spatial Comparisons Across Years. The comparison of spatial patterns for specific days in different years (e.g., Fig. X) does not convincingly illustrate trends or model skill. Given interannual variability in meteorological conditions, such snapshots may not be representative. A more statistically robust analysis of temporal trends or climatological comparisons is needed.

The reasoning for showing certain days is that the background of the paper is in frost quake research and for this reason maps on the 6th of January 2016 and 2023 are shown for SWE and soil T, which are dates when frost quakes have been observed in other papers. The maps are there to visualize some of the outputs from HydroBlocks. The statistical analysis on model performance is included in tables 3-6. The focus of the paper is not to do analysis of trends.

1. Title and Focus Mismatch. The title promises an analysis of "temporal and spatial trends," but the manuscript lacks a dedicated trend analysis. The results are largely descriptive and do not systematically quantify or discuss long-term changes in soil conditions. This disconnect should be addressed.

Here, the word trend is meant to describe how, for example, the temperature profile behaves across time. Trend analysis was never the purpose. We realize now that the wording is misleading. Using wording such as "soil dynamics" would be better. The paper would benefit from a title with better working, and we suggest something along the following: "High resolution modelling of temporal and spatial dynamics in soil conditions in subarctic Finland using HydroBlocks model".

1. Redundant Figures. Figures 10, 11, and 12 contain repeated subplots of meteorological and SWE time series, which adds little value and distracts from the key messages. These could be consolidated or moved to supplementary material.

The reason for leaving input air temperature and modelled snow water equivalent in the plots with soil temperature and soil water content is to show how the atmospheric and snow conditions directly relate to the modelled soil conditions. We agree that to save space, these figures should be combined to present air temperature, SWE, soil T and SWC only once in the same figure to reduce repetition.

1. Weak Scientific Narrative. The manuscript reads more like a technical report than a cohesive scientific paper. The focus is diffuse, shifting between frost quakes, model validation, and regional climatology without a clear central question. I recommend reframing the study to emphasize either (a) advancements in land surface modeling at high resolution, or (b) a focused investigation of frost quake drivers using HydroBlocks. The current version would benefit from substantial restructuring and refocusing before reconsideration.

The focus of the paper is in finding a way to estimate soil temperature and soil ice content with high spatial resolution across the multiple decades, to be able to, in the future, calculate thermal stress in Finland in a similar scale. We think this focus will become clearer after making changes suggested by the reviewers.

---

## Author Comment (AC2)

Thank you to the reviewers for your feedback and suggestions. Please find the response to your comments in blue.

**Reviewer #2**

The paper "Modelling of temporal and spatial trends in soil conditions in Finland using HydroBlocks model" uses outputs from a land surface model to estimate thermal stress in frozen soil to analyze the co-occurrence with frost quakes in Finland. The analysis includes comparing model outputs of snow water equivalence, soil water content, soil temperature and soil ice content between the model and observations for several sites in Finland. The paper is well written and has a logical organization that makes it very comprehensible. The subject matter is interesting and provides a unique application for land surface model outputs. The paper can be approved by addressing inconsistency in terminology, revising several plots, and strengthening the conclusions. Specific comments for revisions are given below.

Revisions:

Line 48: There is a lot of text devoted to the description of the thermal stress and even including an equation for thermal stress. This made me think that the analysis would be analyzing thermal stress in the soils. However, this paper is really about validating a land surface model for the potential use of estimating thermal stress. It could be helpful to cut back some of the text describing the thermal stress and focus more on how land surface models estimate the key inputs needed for thermal stress.

We agree with this comment that it is not necessary to go so deeply into describing thermal stress at this point. The reason for including the equation is to establish a connection between thermal stress calculations and using a land surface model; the need to obtain temperature distribution in the soil and frost depth. The overly detailed description of frost quakes will be removed from this paper.

Line 67 – Here you first introduce soil moisture content (SMC) and soil water content (SWC). It is not clear what the difference is here and is likely a model specific designation and sometimes the two are used interchangeably in broader hydrology. The readability of the paper would improve if you define specifically what you mean by SMC and SWC and how the two are fundamentally different. Also, from later analysis it is clear that what you really want is soil ice content, so it may make sense to start with that and describe that first.

We agree that the paper would benefit from a clearer description of these. The model outputs the total soil moisture content (SMC), which comprehends both the liquid and frozen water in the soil. It also outputs the soil water content (SWC), which is the liquid part of the soil moisture, and inherently there is also the soil ice content (SIC) which is the part of soil moisture that is in frozen form. For frost quake research, it is true that we want the soil ice content. The reason why we are focused on the soil water content, is the observational soil stations which, in addition to soil temperature, measure soil water content. We will make the distinction clearer by only focusing on SWC and SIC, which are relevant for this paper.

Line 72: The word paring "multiple different" is redundant and should be changed to either "multiple" or "different".

Line 78: "soil water equivalent" should be "snow water equivalent".

Line 322: To help avoid confusion, make the column headers Table 4 consistent with how they are described in the paper and the caption (i.e. SWC-5cm, not W5cm). Same for Table 5 and 6.

*Thank you for pointing these out, the wording of these three points on lines 72, 78 and 322 should and will be changed accordingly.*

Line 105: It is unclear what the sentence that starts with "The boundaries of the subdomains" is trying to communicate. It would be helpful if this was revised.

*We agree this can be difficult to understand solely in a written context. To make this more understandable, the next sentence in line 105 will be followed by an additional sentence and reference to figure 2b.*

*…The boundaries of the subdomains are adjusted in a way that no watershed is split between polygons, and they only belong to the polygon where they fall the most (Figure 2b). This becomes more evident in figure 2b, showing the clusters of watersheds, where the watersheds defined by the model exceed the boundary (red square) of the subdomain.*

Line 276: Figure 7 adds very little information that cannot be assessed from Table 3. In particular, the scale of the x-axis is such so that it extremely difficult to see differences between the models and the observations. One way to improve this is to show a short time period for the gauge that does the best (Vaala) and the gauge that does the worst (Konnevesi). This will give the reader more insight as to why the model performs well at one location and not so well at another.

*This is a great comment. Showing additional shorter time periods for the best and the worst SWE stations would also help to see the effect of snow cover melt time to soil temperatures. The reason for using longer timescales was to show the amount of observational data and modelled decades to better see the variability between years and stations. The paper will benefit from showing shorter time periods as well.*

Line 326: As mentioned above, the description of SMC, SWC, and SIC is very confusing. This seems to be more of nuance from the specific modeling framework. It would be clearer if in the methods section you say that you use SMC and SWC from the model to get SIC. Then just treat SIC as another model output. Given that most of the analysis is focused on SWC, then you can remove SMC from the paper entirely. This could help clean up the terminology and make the analysis clearer.

*Yes, we agree that this terminology is overly complicated and SMC could be removed completely, leaving only the liquid and frozen water contents to discuss.*

Line 392: Conclusion 5 is not really a conclusion. It just uses two outputs from the model to create an output that the model doesn't current output.

*Following the above suggestion of removing SMC and describing only SWC and SIC, this will be re-worded to focus on how the model gives us a country wide estimation of frost depth over the last two decades.*

*5. Additionally, SIC gives estimation on the frost depth and its spatial coverage across Finland, hence giving information on winter soil conditions over the past decade.*

Line 394: Conclusion 6 is too broad for what is analyzed in this paper as there was no analysis of thermal stress or predictability and risk of frost quakes. This paper did validate the inputs need to calculate the

thermal stress and demonstrates that there could be potential for calculating thermal stress and frost quakes, but it does not actually validating the model against observed thermal stress and predicting frost quakes. Rewording this to focus on what was shown in the results would be beneficial.

The sentence in the line 394 should be reworded for example as follows:

6. Furthermore, the outputs from HydroBlocks, soil T and SWC validated against observations and SIC, show potential to be used to calculate thermal stresses following Okkonen et al. (2020) in soils across Finland and further to estimate the risk of frost quakes.